# Preliminary Study of Efficacy and Safety of Self-Administered Virtual Exposure Therapy for Social Anxiety Disorder vs. Cognitive-Behavioral Therapy

**DOI:** 10.3390/brainsci12091236

**Published:** 2022-09-13

**Authors:** Izabela Stefaniak, Krzysztof Hanusz, Paweł Mierzejewski, Przemysław Bieńkowski, Tadeusz Parnowski, Sławomir Murawiec

**Affiliations:** 1First Department of Psychiatry Institute of Psychiatry and Neurology, 02-957 Warsaw, Poland; 2Institute of Psychology, Polish Academy of Sciences, 00-378 Warsaw, Poland; 3Department of Pharmacology, Institute of Psychiatry and Neurology, 02-957 Warsaw, Poland; 4Department of Psychiatry Medical, University of Warsaw, 02-353 Warsaw, Poland; 5Polish Psychogeriatric Society, Polish Academy of Sciences, 00-901 Warsaw, Poland; 6Dialogue Therapy, 02-791 Warsaw, Poland

**Keywords:** social anxiety disorder, virtual reality exposure, cognitive-behavioral therapy, self-administered therapy, exposure

## Abstract

Social anxiety disorder (SAD) is one of the most frequent mental disorders. Exposure to virtual reality can be a solution complementing standard CBT (cognitive-behavioral therapy) or can be used as an independent therapeutic tool. The study’s objective was to assess the safety and efficacy of using self-administered virtual reality exposure vs. CBT and CBT with virtual exposure. We assessed the efficacy of the applied intervention with the Leibowitz Social Anxiety Scale (LSAS). We compared three groups: CBT (*n* = 25), CBT + VR (*n* = 29), and self-administered therapy without aid of a therapist (*n* = 19). The results indicated that all three groups showed changes on the LSAS. The simple effect analysis showed that there were no differences between experimental conditions at T0 (session 1) and T1 (session 9) and that the only significant difference occurred at T2 (session 14). The pairwise comparisons showed that the participants in the VR condition scored higher on the LSAS score during the measurement at T2 than participants in CBT condition. Our study has several limitations. The presented initial study shows that the methods of CBT for social anxiety used so far are also effective, while the VR tool for self-therapy requires further research.

## 1. Introduction

Social anxiety disorder (SAD) is one of the most frequent mental disorders, with 7–9% of the general population being affected [1]. It most often starts in early adulthood and has a chronic course [2]. The vast majority of people report symptoms before 18 years of age [3,4]. SAD is characterized by a marked and persistent fear of one or more social situations. The anxiety is related to the fear of being criticized. The fear in this situation is disproportionate to the real threat posed by the situation. Patients often avoid social stimuli or experience severe anxiety. To be diagnosed with social anxiety, a patient should report the symptoms for longer than 6 months [5] The disorder is associated with impairment in vital domains of daily life, such as occupational/academic and family functioning, relationships, and social activities [6]. Individuals with social phobia, particularly those with the generalized subtype, often show a high degree of comorbidity with other anxiety and affective disorders [7] as well as with alcohol abuse [8]. Usually, social phobia precedes the onset of comorbid conditions. It is a serious social and clinical problem, which implies that people experiencing its symptoms also have the feeling of lower quality of life [9]. Cognitive-behavioral therapy (CBT) is a proven and well-documented treatment for social phobia [10,11,12]. Following the early formulation of cognitive models for social anxiety disorder by Clark and Wells [13] and Rapee and Heimberg [14], a wide range of CBT protocols have been developed. Though most of these have proven effective, the most researched treatment is a combination of exposure and cognitive restructuring [15]. Treatment approaches to SAD include cognitive behavior therapy [16,17], exposure group therapy [18], in vivo exposure therapy [19], and recently virtual reality exposure therapy (VRET) [20]. The use of in vivo exposure is based on models of fear development that implicate the learned nature of particular fears and the instrumental role that avoidance plays in maintaining anxiety. In the exposure treatment of social anxiety disorder, the patient develops an exposure hierarchy, or a list of feared situations, which ranges from situations provoking moderate to extreme anxiety. Using this hierarchy, patients are encouraged to systematically expose themselves to their feared situations and to stay in the situation until their anxiety has subsided [21].

Several published meta-analyses have examined CBT for social anxiety disorder. They compared exposure plus cognitive restructuring with exposure-only treatments for SAD. In this meta-analysis, the two types of CBT were similarly effective, but a higher number of exposure sessions was related to a better outcome [22,23]. The most recent meta-analysis of CBT for SAD included 32 RCTs with a pooled total of 1479 participants. The authors found that CBT produced better posttreatment outcomes than wait-list, psychological placebo, or pill placebo [24]. There are three approaches to implementing exposure therapy in social anxiety disorders: in vivo exposure, imagery exposure, and virtual reality exposure.

Virtual reality (VR) has become an interesting alternative for the treatment of SAD and can constitute an alternative to in vivo and imagery exposure. VR provides a human–computer interaction that allows patients to feel a sense of presence and immersion in a virtual environment, offering an opportunity to expose clinically anxious individuals to realistic life scenarios, thereby reducing their reactivity to anxiety-provoking cues. The impact of VR technologies is discussed in many studies and meta-analyses [25,26,27,28]. Powers et al. were the first to demonstrate in a healthy sample that a virtual reality conversation task led to a similar increase in feelings of anxiety in participants as an in vivo conversation task [29]. For this and many other reasons, the virtual social environment is being used more and more often. VR exposure can be less time-consuming, requires less work in the organization of an exposure situation, and gives the therapist control over the context and intensity of the exposure [30]. Research results are divergent as to whether virtual exposure is as effective as in vivo exposure [31]. The Chesham meta-analysis reported a significant effect of VRET for SAD in comparison to the waiting list and no difference between VRET and CBT [32]. Wechsler et al. published a meta-analysis on RCTs, specifically comparing the efficacy of VRET to in vivo exposure in anxiety disorders [33]. The comparison revealed a small but nonsignificant effect size favoring in vivo exposure. However, some results suggest the opposite. For example, the study conducted by Bouchard et al., which involved two active conditions (VR and in vivo exposure) and a control group (waiting list), showed that VR exposure was more effective and that the treatment effects were still measurable during the follow-up six months after the completion of the study [34]. The main limitation of the study was the small size of the groups. Thus, assessing the efficacy of virtual exposure compared to in vivo exposure requires further investigation.

The patient’s involvement in the virtual exposure, and thus the impact of this reality on the habituation process, depends on many factors, including those related to the technological advancement of the virtual environment and factors related to the patients themselves [35]. In the literature to date, there are preliminary studies on the use of therapeutic programs with virtual reality but without the aid of an actual therapist in the treatment of social phobia [36]. Those studies were primarily exploratory research. In one such study, there was a system of home autotherapy in which patients used virtual reality exposure and the support of an e-therapist [37]. Such methods can be treated as innovative approaches to therapy requiring additional efficacy evaluation in clinical studies. The various benefits patients can derive from social exposure in virtual reality as autotherapy are very interesting and helpful in therapeutic practice. We would like to explore the safety, efficacy, and usefulness of the tool for self-administered VR exposure compared to the standard therapeutic approach to a social anxiety disorder.

The aim of this study was to evaluate the effectiveness and safety of the self-exposure tool in virtual reality in patients diagnosed with social phobia in comparison to active groups (CBT and CBT + VR). The efficacy was assessed using the Leibowitz Anxiety Scale, and the safety assessment focused on the occurrence of simulator disease symptoms during exposure. The self-administered VR exposure is a tool that is operated by the patient without the support of a therapist.

## 2. Design of the Study

The study was a randomized, open-label, single-blind, controlled trial. The study was preregistered at clinicaltrial.gov under number NCT03895957 and was approved by the Bioethics Committee of the Regional Medical Chamber in Warsaw (KB/1214/19). In the study, we decided to compare the efficacy of the therapeutic effects in three parallel groups:1A self-administered VR group (experimental group), where the patients were gradually exposed to social situations in virtual reality. The patients had several exposures at their disposal in VR, which they selected themselves, taking into account the severity of the anxiety. The patients underwent the therapy without any kind of help or intervention from a therapist.2A CBT + VR group (experimental group), where virtual reality was used in the CBT protocol. Within this arm, a therapy analogous to that which took place in the CBT group was carried out, while the exposure in virtual reality replaced the exposure in the patient’s imagination.3A CBT group (active control group) in which work was based on a cognitive-behavioral therapy protocol.

Regarding the purposes of the therapy, a therapeutic protocol was prepared, on the basis of which cognitive-behavioral therapists conducted their sessions. The therapy was based on the Clark and Wells model (1995). In this arm, the protocol assumes that exposure to social situations takes place in the patient’s imagination.

### 2.1. Participants

We planned our sample size based on the results of Yoshinaga et al. [38]. We estimated an effect size of 30 points (SD = 30) on the Liebowitz Social Anxiety Scale [39], our main efficacy measure. The results of a power analysis for a repeated-measure analysis of variance conducted in G*Power (v. 3.1; *f* = 0.4, *α* = 0.05, *β* = 0.8) showed that we had an 81% chance of correctly rejecting the H0 of no significant effect of the interaction (time vs. group) with a total sample of 78 participants (26 participants per group). We assumed a 15% dropout rate and estimated our required sample size to be 30 participants per condition.

All participants were Polish. The participants were recruited via social media, e-mails, and a dedicated website (www.tomorrow.pro/vrmind (accessed on 12 August 2022), and they were then qualified for the study on the basis of the inclusion and exclusion criteria. The inclusion criteria were: age (18–50 years old), confirmed SAD (diagnosis via DSM IV-TR criteria) lasting for at least two years, stable pharmacological treatment—no change in pharmacological therapy during the three months prior to the study, and signed informed consent. The exclusion criteria were: psychoses, bipolar affective disorder, mental disability, pregnancy, addictions, attending a therapeutic session under the influence of alcohol, (ongoing) treatment by a neurologist for chronic CNS disease, epilepsy, seizure dizziness, the presence of (current) suicidal thoughts, tendencies, or attempts, and currently undergoing CBT therapy. The participants were randomly assigned to each group, CBT, CBT + VR, and VR, based on block randomization. Randomization took place using computer software. Randomization was carried out after the end of the visit (T0), consisting of a mental condition assessment. See Figure 1 for an overview of the randomization procedure, and for the sample characteristics per condition, see Table 1. Thirty-nine men and fifty-two women participated in the study. There were no age differences within our sample based on gender (*F* (1,84) =0.13, *p* = 0.71, *ŋ_p_*^2^ = 0.00), experimental condition (*F* (2,84) = 0.04, *p* = 0.95, *ŋ_p_*^2^ = 0.00), or the interaction of these factors (*F* (2,84) = 2.55, *p* = 0.08, *η_p_*^2^ = 0.05) (one person from the VR arm did not provide information about age). The proportion of gender did not differ by condition (*χ*^2^ = (2, N = 91) = 0.01, *p* = 0.99).

### 2.2. Measures

We used a Polish adaptation of the Structured Clinical Interview for Axis I Disorders DSM-IV-TR [40] as the main screening tool for confirming SAD. SCID-I is a semi-structured clinical interview aimed at diagnosing Axis I disorders according to the DSM classification-IV-TR.

We used the Clinical Global Impression Scale (CGI) [41] to assess the severity of SAD. CGI is a clinician-completed, single-item, seven-point scale used to evaluate the severity of illness, where one is labelled as “Normal, not at all ill” and seven is labelled as “Among the most extremely ill patients”. In addition to the CGI, we also used the Patient Global Impression Scale (PGI), where the patient defines his subjective experience related to his current well-being and functioning (where one is labelled as “Normal” and four is labelled as ”Severe”).

The Liebowitz Social Anxiety Scale [39] was used as the primary outcome measure. The scale is composed of 24 items depicting various social situations. For each item, participants assess their fear (from 1, —“No fear”, to 4, “Severe”) and avoidance (from 1, “Never”, to 4, “Usually”). The LSAS has three scores, summing up the results for the particular items: fear (0–72), avoidance (0–72), and the total score (0–144). The internal consistency of this scale in our study was excellent (Cronbach’s *α* = 0.87–0.97).

The widely used Beck Depression Inventory (BDI) [42], Clinician Global Impressions of Improvement (CGI-I), and Patient Global Impression of Change Scale (PGICS) [41] were applied as secondary outcome measures. CGI-I is a clinician-completed scale that measures the change in the severity of symptoms, ranging from 1, labelled as “Very much improved”, to 7, labelled as “Very much worse”. The PGICS is a patient-completed scale that measures the change in the severity of symptoms, where one is labelled as “No change” and seven is labelled as ”Very much improved”. Finally, the BDI is a 21-question multiple-choice self-report inventory for measuring the severity of depression. Each answer is scored on a scale value of 0 to 3. A higher total score indicates more severe depressive symptoms. In our sample, the BDI reliability was excellent (Cronbach’s *α* = 0.88–0.93).

To assess the severity and occurrence of simulator sickness, we used the Simulator Sickness Questionnaire (SSQ) [43]. The questionnaire contains 26 items. Participants assess the occurrence and severity of each symptom with four labels (0—“none”, 1—“slight”, 2—“moderate”, and 3—“severe”). Individual scores were corrected for baseline (pre-VR) symptom severity. Only items with a “post-VR—pre-VR” increase contributed to the final SSQ score. A total SSQ score equal to or higher than ten was used as a preliminary cut-off for simulator sickness [44].

In addition to the above, we gathered information about the use of VR systems. The computer system recorded information about each VR scenario, including its duration and the participant’s speaking time (active input on a microphone during periods when participants were asked to speak). The VR system asked the participant to assess his/her subjective units of distress (SUD; 0–100) before and after each exposure (after the exposure, participants provided feedback on their actual state as well as on the maximal perceived SUD during the exposure). SUD is a subjective measure of perceived fear in a certain situation. The scale range is from 0 (“Totally relaxed) to 100 (“Highest distress/fear/anxiety/discomfort that you have ever felt”). The scales used in the study are presented in Table 2.

### 2.3. Treatment

The study used proprietary software called VR Mind^TM^ and VR Mind+^TM^, which uses a virtual helmet. The software consists of nine therapeutic scenarios (job interview, public speaking in an auditorium, speaking at a meeting in a conference room, purchasing a ticket at a railway station, restaurant visit, telephone call in a public place, train compartment, returning goods in a shop, and a social call) and a training scenario (learning controls, movement, etc.). Each therapeutic scenario has three levels of difficulty.

In the study in the VR and CBT + VR arm, the following equipment was used: HTC VIVE VR goggles with 6 degrees of freedom (6-DOF), a viewing angle of over 100 degrees, a viewing width of 110 degrees, and a refresh frequency of 90 Hz. The computer used included a gtx 960/8 GBRAM card, Windows 10, a 256 GB disk, and a motherboard compatible with an Intel i7 9600K processor.

The therapy in the CBT and CBT + VR group was conducted by certified cognitive behavioral therapists.

Prior to the examination, the therapists received 10 h of training on the manner in which to conduct the therapy based on the protocol. Then, during the entire time of working with the patients, they performed supervisions (2x a month, 4 h each), and their task was to assess the compliance of the session with the protocol. During the study, the monitoring of events/side effects of the conducted interventions was carried out (CBT, CBT + VR, and VR). In the CBT control group and experimental CBT + VR group, the intervention was based on a protocol created on the basis of the Clark and Wells model, and therapeutic sessions were held twice a week, each lasting 45 min (slightly shorter than a standard CBT session). In total, the patients had 12 therapeutic sessions (Table 3).

Due to the shorter duration of the therapy, we decided to omit two typical elements of the therapy present in the CBT model: experiments checking social standards and the use of video recordings during the experiments.

### 2.4. Procedure

After entering the laboratory, the participant was asked to become familiar with basic information about the research and to sign the informed consent. The clinical evaluation was performed by an independent clinician in order to confirm the SAD diagnosis conducted by the SCID and filled-in CGI. In the next step, the participant completed a series of questionnaires in a fixed order: LSAS, BDI, PGI, and SSQ-PRE. The physician did not know which group the patient would be assigned to. After the therapy, the patients were re-examined by the clinician, who was not informed which group the patient belonged to. Then, the participant took part in the first VR training session where he/she became familiar with the VR environment.

Shortly after finishing the training session, the participant filled in the SSQ-POST. At the end, the computer software randomly assigned the participant to one of the conditions. The duration of the first session was 90 min. The subsequent sessions, from 2 to 13, were held according to the session plan of the therapeutic protocol for each arm (Table 2). Each session lasted 45 min. In the CBT + VR and VR arms, the SSQ was administered during each session. In the CBT arm, the SSQ was administered during sessions 7, 8, 10, 11, and 12. After the 9th session (T1), patients were re-examined with the Leibowitz Social Anxiety Scale. The last session (T2) was the final assessment of the effectiveness of the proposed impacts. During this meeting, the patients filled in the following questionnaires: LSAS, BDI, PGI-CS, and CGI-I.

### 2.5. Statistical Analysis

The results were calculated with the jamovi statistical package and the GAMLj library [39]. Due to high dropout rates in the VR group (36%), we decided to switch from a traditional within-subjects ANOVA to mix models, as they are more suitable for longitudinal data with missing values and the unbalanced design of our main efficacy measures. We used the Satterthwaite method to estimate the degrees of freedom. The dependent variables followed normal distributions as did the model residuals [45].

## 3. Results

### 3.1. T0—The Severity of SAD Symptoms

Before the main analysis, we checked if there were any differences at T0 between the conditions in the severity of SAD symptoms and the participants’ everyday functions. The analysis of variance showed no differences in the CGI and PGI scores (see Table 4 for an overview of the comparisons and descriptive statistics).

### 3.2. Primary Endpoints

#### 3.2.1. Efficacy

We built a mixed model with time of measurement as a fixed factor and a random intercept for the total LSAS score using the Satterthwaite method for degrees of freedom. The model revealed a significant effect of session (*F*(2,149.31) = 54.71, *p*
*<* 0.001). The estimated coefficient showed a reduction of −23.73 (95% CI: −28.23 to −19.23) points in the total LSAS score between the first and the last sessions.

In the second step, we added an experimental condition as a fixed factor, and in the third step we allowed for the interaction of both variables. The main effect of condition was nonsignificant (*F*(2,87.96) = 1.04, *p* = 0.355), but the interaction between time of measurement and condition showed significant differences in the total LSAS score (*F*(4,144.9) = 2.65, *p* = 0.035). The significant interaction, jointly with the ratios of the marginal and conditional pseudo *R*^2′^s and AIC criterion, supported the model with main effects of condition and time of measurement and their interaction (see Table 5 and Table 6 and Figure 2).

The simple effect analysis showed that there were no differences between experimental conditions at session 1 (*F*(2,122.56) = 0.35, *p* = 0.700) and session 9 (*F*(2,135.32) = 0.21, *p* = 0.808) and that the only significant difference occurred at session 14 (*F*(2,133.9) = 206 4.15, *p* = 0.002). The pairwise comparisons showed that the participants in the VR condition scored higher on the LSAS score during the measurement at session 14 by 19.76 (*SE* = 7.25) points than participants in CBT condition (*t*(153.271) = 2.74, *p* = 0.007). There were no other significant differences between conditions.

In the next step, the LSAS subscales were analyzed to determine the source of differences in the LSAS total score. We built a separate mixed model for each subscale. The anxiety subscale results were similar to the results obtained on the LSAS total score. For the avoidance subscale, the pattern of results was also similar. However, the interaction between time of measurement and condition did not meet the conventional significance threshold (Figure 3).

#### 3.2.2. Safety Assessment

In order to assess safety, we used the SSQ. For each session in which the level of the simulator sickness was measured, the difference between the second and the first SSQ measurement was calculated (measurement II—measurement I). We summed up all positive differences and all values above ten were classified as having simulator sickness [34]. According to the adopted algorithm, out of 973 therapeutic sessions in which the SSQ measurement was performed, there were 11 results indicating the occurrence of symptoms characteristic of simulator sickness. The results indicated that the symptoms of simulator sickness occurred in three people in the VR arm and one person in the CBT + VR arm. Additionally, four people in the CBT condition reported symptoms similar to simulator sickness. In seven out of eight people, the symptoms of simulator sickness appeared once. One person in the VR group experienced four episodes. However, it did not result in withdrawal from the therapy. One male in the VR condition who experienced an episode of simulator sickness in session #3 did not appear in session #4 and withdrew from the study. Due to the low number of cases with simulator sickness symptoms, no further statistical analysis was conducted.

### 3.3. Secondary Endpoint—Efficacy

According to the assumptions of the study, secondary endpoints were also determined, and these results were obtained using Clinical Global Impression Improvement (CGI I), Patient Global Impression of Change (PGIC), and the Beck Depression Inventory (BDI).

In the case of the CGI-I and PCGI measurements, we used a one-way variance model with a three-levelfactor, which is the experimental condition (CBT vs. CBT + VR vs. VR). The results of the ANOVA for CGI-I (1, “Very much improved”, and 7, “Very much worse”) showed a significant effect of the experimental condition (F (2,70) = 6.49, *p* = 0.003, *η_p_*^2^ = 0.15). Pairwise comparisons disclosed significant differences between the CBT (*M* = 2.56, *SD* = 0.71) and VR conditions (*M* = 3.21, *SD* = 0.63; *t*(42) = 3.15, 237 *p**_adj_* = 0.003, *d* = 0.96) and no differences between CBT and CBT + VR (*M* = 2.55, *SD* = 0.68). We obtained a similar pattern of results with the PGICS (*F*(2, 70) = 8.40, *p*
*<* 0.001, *η_p_*^2^ = 0.19). Participants in the CBT condition reported greater improvement (*M* = 5, *SD* = 1.12) than participants in the VR condition (*M* = 3.74, *SD* = 1.37; *t*(42) = 3.37, *p**_adj_* = 0.002, *d* = 1.03). We found no difference between CBT and CBT + VR (*M* = 241 5.1, *SD* = 1.18) (Table 6).

Finally, we checked if the groups differed on the BDI scores before and after the intervention. In order to accomplish this, we built a mixed model for the BDI total score with session as fixed factor (session 1 vs. session 14) and a random intercept. The result showed a significant effect of session (*F*(1, 90) = 56.6, *p*
*<* 0.001), with an intercept estimate equal to 13.2 (95% CI: from 11.36 to 15.03) and estimate reduction of −7.5 points (95% CI: from −9.54 to −5.06) between session 1 and session 14. Adding group and the group and session interaction did not reveal any other significant effects. This means that the BDI total score reduction was comparable within experimental conditions.

### 3.4. Other Analyses

#### 3.4.1. Patient Dropouts

Overall, 18 out of 91 participants did not complete all 14 sessions (5 in CBT, 2 in CBT + VR, and 11 in VR). The chi-squared test results showed that this difference was significant (*χ*^2^ (2, N = 91) = 9.04, *p* = 0.01).

Specifically, the dropout rate in the VR condition was significantly higher than in the CBT + VR condition (*χ*^2^ (1, 254 N = 61) = 6.7, *p* = 0.01) and simultaneously there was no difference between CBT and VR (*χ*^2^ (1, N = 60) 255 = 2.13, *p* = 0.14).

#### 3.4.2. VR System

In the VR condition, 30 participants launched a total of 654 scenarios. The mean time of a single exposure was 7.42 min. The three most frequently selected scenarios were: public speaking in the auditorium (112), job interview (88), and buying a ticket at a train station (68). The mean speaking time was 1.65 min. The average subjective units of distress were equal to 18.12 before exposure, 18.37 after exposure, and 23.63 at maximum during the exposure. In comparison, in CBT + VR, 31 participants launched a total of 184 scenarios. The mean time of VR exposure was 7.85 min. The most frequent scenarios were public speaking in the auditorium (52), job interview (32), small talk meeting (28), and speaking in a conference room. The average subjective units of distress were equal to 29.70 before exposure, 28.60 after exposure, and 40.65 at maximum during the exposure.

## 4. Discussion

Studies on the use of virtual reality in patients with social phobia are more and more common, and their number has recently increased. The available meta-analyses are ambiguous. One of them indicates that VRET (virtual exposure therapy) is an acceptable method of social phobia therapy and brings a long-lasting effect. The long-term effectiveness of VRET may decrease compared to in vivo and in sensu exposure [46]. In his review, Emmelkamp indicated that there are no significant differences between CBT and VRET, but there is a need for further studies to assess the effectiveness of VRET as a stand-alone therapy [20]. On the other hand, the Chesham study shows that there are no differences between VRET and in vivo and in sensu exposure, which makes these types of exposure equivalent in the treatment of patients with social phobia. The study by Zainal et al. evaluated the effectiveness of VRET vs. the waiting list, and this study favors VRET [47]. Pellisolo’s study shows that VRET can be seen as an alternative to CBT and SSRI therapy [48].

In the described study, the results confirmed the lower effectiveness of the self-administered VR therapy compared to the standard CBT and CBT + VR therapy. Similar to Emmelkamp’s postulate, we believe that the autotherapy tool, VR, requires an assessment of efficacy against an inactive control group and then an analysis of the result and an evaluation of the strengths and weaknesses of the tool. Another important issue is the use of exposure as part of standard CBT therapy. Research indicates that many therapists do not have full control over the conducted exposure, whether in sensu or real life, and the intensity of the exposure may be insufficient [49,50]. The therapist does not know whether the patient is practicing the exposure at the time of performing the exercise and cannot fully assess its intensity or the intensity of the emotions experienced by the patient. In addition, many studies show that exposure exercises are not chosen by the therapist as frequently as they should due to the difficulty in their planning and implementation [51]. Virtual reality allows therapists to be more precise in planning social exposure in terms of the number of people in the virtual reality or the difficulty of the given social situation, which enables them to make use of the interactive nature of the exercise. It is also possible to analyze the level of anxiety through a system of data collection during VR exposure.

A major limitation in the described study is the lack of assessment of the patients’ “sense of presence” in virtual reality. “Sense of presence” is defined as the interpretation of the artificial environment as if it were real. Many studies view this construct as a mechanism that can make VR an effective tool for exposure therapy [52]. This parameter is one of the factors that affects the extent to which the patient is able to expose himself or herself to virtual scenes. Information from the observation of the patients’ reactions in our study has led us to the conclusion that some patients did not feel a sufficient sense of presence in the displayed scenes. However, this aspect requires further investigation.

Patients in the VR self-administered therapy group were more likely to drop out of the intervention prematurely than those in other groups. This prevented us from achieving the research goal of analyzing 30 individuals. At the initial planning stage of the study, we did not consider such a high dropout rate in this study condition. Our tentative hypotheses assumed: (a) a lack of motivation to stay in the situation of exposure, which may be linked to a lack of therapeutic alliance, (b) excessive anxiety triggered by the scenes, (c) difficulties in overcoming avoidance, and (d) a low level of sense of presence in VR. Dropouts in the VR self-administered therapy group were not associated with aggravated simulator sickness.

The study showed that the use of virtual reality exposure using the HTC VIVE virtual helmet is safe. Symptoms of simulator sickness occurred only in a small number of cases and were of low to moderate severity, which did not require interruption of the exposure. What is interesting is the false-positive results recorded in the pre- and post-SSQ questionnaires in the CBT group. These results can be interpreted in two ways. On the one hand, it is probable that the applied SSQ scale does not measure the symptoms of simulator sickness accurately enough, and physiological data should be additionally taken into account [53]. On the other hand, the symptoms of anxiety during the exposure could have been interpreted by the patients as symptoms of simulator sickness. Based on these false-positive results, one might be tempted to conclude that the applied scale does not objectively measure the severity of the symptoms of simulator sickness. A recent meta-analysis presented similar conclusions [54]. When interpreting the total SSQ scores, according to Kennedy et al. [43], scores between 10 and 15 indicate significant symptoms, and scores from 15 to 20 indicate severe symptoms, while scores above 20 are indicative of simulator sickness. It should, however, be stressed that the given values were established for military personnel using flight simulators, and the scores may vary in the population at large. Furthermore, SSQ scores tend to be higher in other virtual environments compared to flight simulators [43,55]. Since the SSQ is a self-report scale, participants may not always faithfully reflect the severity of a given symptom. Taking into account all the shortcomings of the scale used, in the VR condition only three people from the VR self-administered therapy group reported symptoms of simulator sickness (one of them confirming the symptoms on four occasions), while in the CBT + VR condition only one participant reported symptoms. This indicates a low severity of symptoms of simulator sickness and confirms the safety of the system used.

Future research should assess the effectiveness of the VR autotherapy tool compared to the inactive control group in order to verify the effectiveness of the method. We assume that the tool could be used by people who have problems reaching a psychotherapist. Moreover, an important direction of research would be the identification of people who could benefit from such a tool in a special way. Another interesting issue that requires further research and deeper reflection is the high dropout rate in the group treated with the autotherapy tool. The project requires further development in order to improve the technical parameters of the virtual reality.

### Limitations of the Study

There are three main limitations of this study. The first is the lack of follow-up. It is not clear how robust and long-lasting the therapeutic effects are or if there are any differences across the conditions. Second, we did not screen participants for the dropout motives, and consequently our results could be biased in any direction. For example, it is possible that our participants discontinued the therapy as a consequence of lack of motivation and positive reinforcement from the therapist, low perceived therapy efficacy, or due to reaching a certain level of habituation (i.e., they were no longer perceiving anxiety during VR exposition). In future studies, detailed analyses of dropout motives should be applied. Finally, our sample size was limited to a total of 91 participants. Future research would benefit from larger sample sizes.

## 5. Conclusions

The results of the presented study indicate that in the three research groups there was a change according to the Leibowitz Social Anxiety Scale (primary outcome) between the measurement points T0 and T2. The worst effect in this respect was obtained in the self-administered VR group, while the results in the other two groups (CBT and CBT + VR) were comparable. Due to the insufficient size of the studied groups and the fact that the research is burdened with insufficient statistical power, we treat the results as a trend that should be studied further. The effects in terms of the LSAS avoidance subscale were similar. The VR self-administered therapy group obtained significantly higher scores on the anxiety level, as measured by the LSAS anxiety subscale. The compared types of therapy can therefore be considered effective in our study group, and the change in exposure type (virtual reality vs. in sensu) did not significantly affect the effectiveness of the intervention in the CBT vs. CBT + VR conditions. The results described in this paper indicate that the use of virtual reality as part of CBT therapy does not significantly affect its efficacy, although it does affect some of its aspects.

## Figures and Tables

**Figure 1 brainsci-12-01236-f001:**
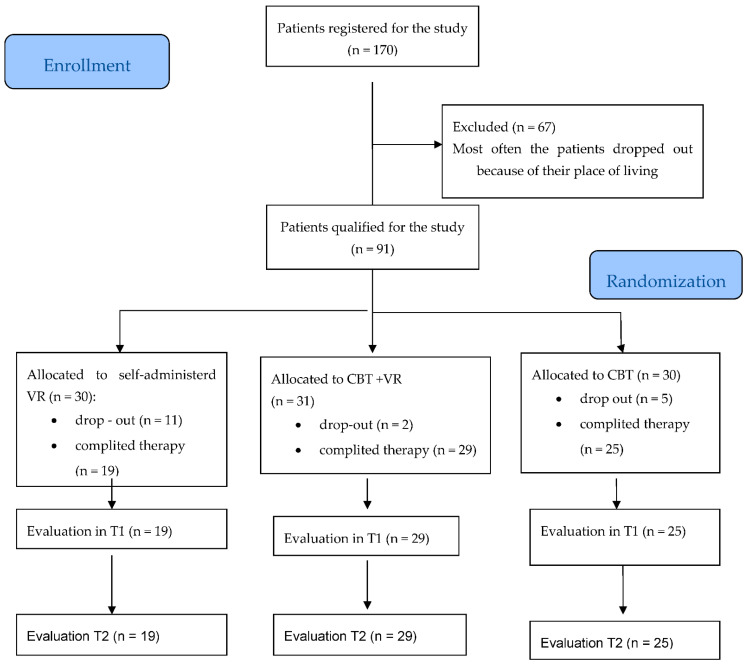
Participant flowchart.

**Figure 2 brainsci-12-01236-f002:**
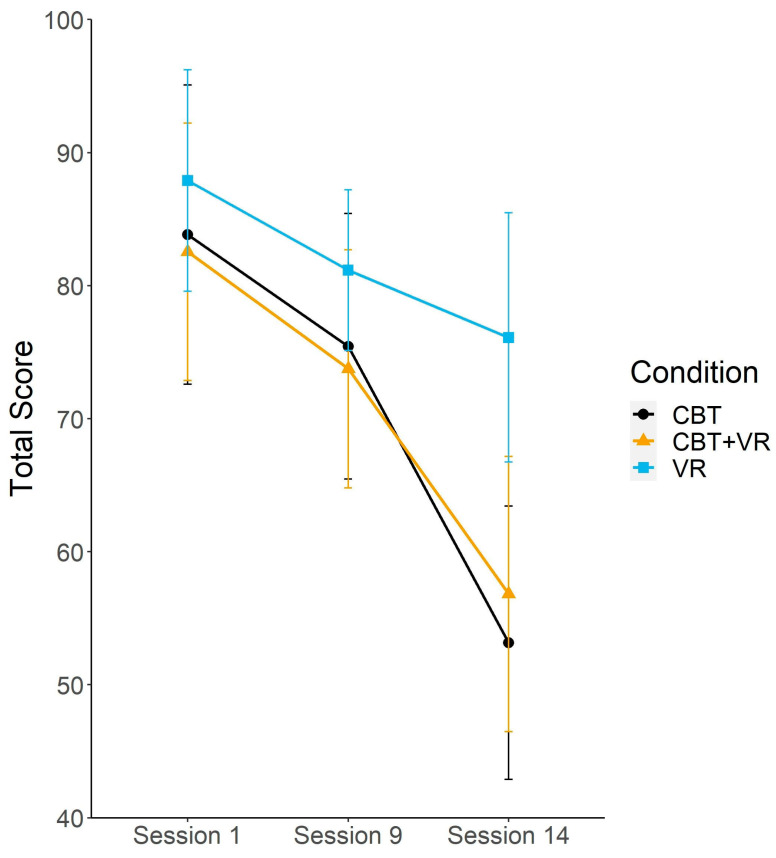
Liebowitz Social Anxiety Scale total score split by condition and measurement time. Error bars represent 95% confidence intervals.

**Figure 3 brainsci-12-01236-f003:**
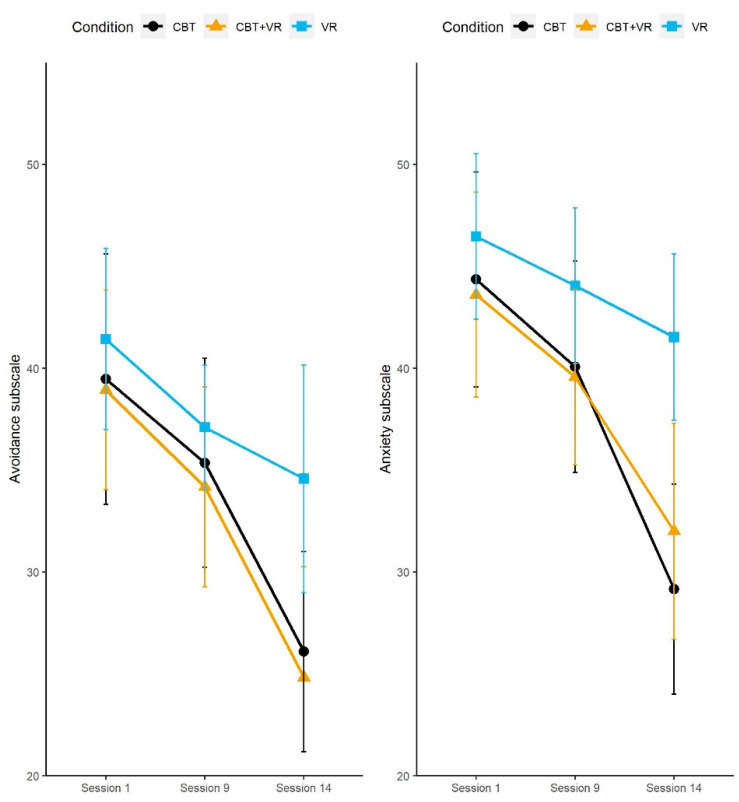
Anxiety and Avoidance subscales of Liebowitz Social Anxiety Scale scores split by condition and measurement time. Error bars represent 95% confidence intervals.

**Table 1 brainsci-12-01236-t001:** Participant characteristics.

Characteristics	CBT	CBT + VR	Self-Administered VR -Exposure
N	30	31	30
Age, *M*(*SD*)	31.44 (7.68)	31.98 (7.06)	31.39 (7.73)
Gender, *n* female (%)	17 (56)	18 (58)	17 (56)
Past treatment, *n* (%) none	9 (30)	13 (42)	8(26)
Pharmacotherapy	14 (46)	11 (35)	18 (60)
Psychotherapy	20 (66)	22 (70)	24 (80)
Both	12 (43)	8 (25)	18 (60)
Current pharmacotherapy, *n* (%)	8 (26)	8 (25)	11 (36)
Session completed, *n* 1	30	31	30
2	30	31	30
3	30	30	26
4	28	30	22
5	28	30	21
6	28	30	20
7	28	30	19
8	28	30	19
9	26	29	19
10	26	29	19
11	25	29	19
12	25	29	19
13	25	29	19
14	25	29	19
Dropouts (%)	5 (16)	2 (6)	11 (36)

**Table 2 brainsci-12-01236-t002:** Measures used in the study.

Scales			
Structured Clinical Interview for Axis I Disorders DSM-IV-TR (SCID-I)	Clinician	Confirming social anxiety disorder	T0
Liebowitz Social Anxiety Scale (LSAS)	Patient	Primary outcome	T0, T1,T2
Clinical Global Impression Scale (CGI)	Clinician	Severity of SAD symptoms	T0
Patient Global Impression Scale (PGI)	Patient	Severity of SAD symptoms	T0
Beck Depression Inventory (BDI)	Patient	Secondary outcome	T0, T2
Clinician Global Impressions of Improvement (CGI-I)	Clinician	Secondary outcome	T0,T2
Patient Global Impression of Change Scale (PGI-CS)	Patient	Secondary outcome	T0, T2
Simulator Sickness Questionnaire (SSQ)-pre	Patient	Before each exposure session	
Simulator Sickness Questionnaire (SSQ)-post	Patient	After each exposure session	

T0—assessment point before intervention, T1—assessment point in 9th session, T2—assessment point after intervention.

**Table 3 brainsci-12-01236-t003:** Short protocols for CBT, CBT + VR, and self-administered VR therapy.

Session (Duration: 45 Min. Each Arm)	Therapy Protocol: CBT	Therapy Protocol: CBT + VR	Therapy Protocol: Self-Administered VR Exposure
2	Presentation of general information about the therapy Identification of therapy goals and problems. Signing of a therapy contract. Homework	Presentation of general information about the therapy Identification of therapy goals and problems. Signing of a therapy contract. Homework	Turning on the application. Selection of a scenario. Reading the instructions for the scenario VR exposure. Self-reflection
3	Familiarization of the patient with the cognitive model of anxiety Psychoeducation on social anxiety Homework	Familiarization of the patient with the cognitive model of anxiety Psychoeducation on social anxiety Homework	Turning on the application. Analysis of the summary of results from the previous sessions. Selection of a scenario. Reading the instructions for the scenario. VR exposure. Self-reflection
4	Introduction of the cognitive model of social anxiety Homework	Introduction of the cognitive model of social anxiety Homework	See above
5	Psychoeducation on cognitive distortions. Introduction of cognitive restructuring as part of discussion with negative automatic thoughts. Homework	Psychoeducation on cognitive distortions. Introduction of cognitive restructuring as part of discussion with negative automatic thoughts. Homework	See above
6	Introduction to behavioral techniques. Homework	Introduction to behavioral techniques. Homework	See above
7–12	Imagination exposure. Discussion of exposure, discussion of a list of safety behaviors, and discussion of beliefs and perceptions about oneself in social situations. Cognitive reformulation. Homework	Virtual exposure. Discussion of exposure, discussion of a list of safety behaviors, and discussion of beliefs and perceptions about oneself in social situations Cognitive reformulation. Homework	See above
13	Therapy summary	Therapy summary	See above

**Table 4 brainsci-12-01236-t004:** Means, standard deviations, and ANOVA results for CGI and PCGI at T0.

	CBT	CBT + VR	VR	*F*	$\eta_{p}^{2}$	*p*
CGI	4.36 (0.85)	4.22 (0.76)	4.36 (0.96)	0.27 *^a,b^*	0.00	0.764
PGI	3.2 (0.55)	3.26 (0.58)	3.33 (0.66)	0.37 *^a,c^*	0.00	0.692

CGI—Clinical Global Impression Scale; PGI—Patient Global Impression Scale; *a*—*df1* = 2; *b*—*df2* = 88; *c*—*df2* = 87.

**Table 5 brainsci-12-01236-t005:** Means and standard deviations for time of measurement of primary and secondary efficacy outcomes.

		CBT			CBT + VR			VR	
	T0	T1	T2	T0	T1	T2	T0	T1	T2
LSAS-anxiety	44.37 (14.13)	40.08 (13.89)	29.16 (13.79)	43.61 (13.71)	39.57 (11.78)	32 (14.46)	46.47 (10.89)	44.05 (10.23)	41.53 (10.96)
LSAS-avoidance	39.47 (16.48)	35.36 (13.71)	26.09 (13.18)	38.94 (13.36)	34.18 (13.35)	24.83 (14.8)	41.43 (11.9)	37.11 (8.19)	34.58 (14.96)
LSAS-total score	83.83 (30.12)	75.44 (26.74)	53.16 (27.49)	82.55 (26.36)	73.75 (24.42)	56.83 (28.19)	87.9 (22.29)	81.16 (16.19)	76.11 (25.1)
BDI	17.62 (9.96)	-	12.23 (9.6)	14.39 (9.51)	-	8.96 (9.21)	19.83 (10.5)	-	16.17 (12.05)
CGII	-	-	2.56 (0.71)	-	-	2.55 (0.69)	-	-	3.21 (0.63)
PGICS	-	-	5 (1.12)	-	-	5.1 (1.18)	-	-	3.74 (1.37)

Note: LSAS: Leibowitz Social Anxiety Scale; BDI: Beck Depression Inventory; CGII: Clinician Global Impressions of Improvement; PGICS: Patient Global Impression of Change Scale.

**Table 6 brainsci-12-01236-t006:** Mix model parameters for the Liebowitz Social Anxiety Scale.

			Random Intercept + Time			Random Intercept + Time + Condition			Random Intercept + Time + Group		
			*B*	*CI*	*p*	*B*	*CI*	*p*	*B*	*CI*	*p*
Total Score	Intercept		74.1	69.2 to 79	<0.001	74.23	69.34 to 79.12	<0.001	74.45	69.57 to 79.33	<0.001
	Session	9–1	−8.18	−12.7 to −3.65	<0.001	−8.02	−12.56 to −3.5	<0.001	−8.13	−12.62 to −3.36	<0.001
		14–1	−23.73	−28.23 to −19.23	<0.001	−23.57	−28.08 to −19.06	<0.001	−22.78	−27.26 to −18.31	<0.001
	Condition	CBT + VR-CBT				−0.18	−11.98 to −11.61	0.975	0.04	−11.72 to 11.81	0.995
	Session * Condition	VR-CBT				−7.62	−4.45 to 19.71	0.219	8.8	−3.34 to 20.92	0.159
		9–1 * CBT + VR-CBT							−0.83	−11.26 to 9.6	0.876
		14–1 * CBT + VR-CBT							4.81	−5.56 to 15.17	0.365
		9–1 * VR-CBT							−1.52	−12.95 to 9.9	0.793
		14–1 * VR-CBT							15.7	4.27 to 27.12	0.008
	Random parts	AIC	2100.42			2102.28			2099.57		
		*σ* *2*	475.43			472.57			471.41		
		*τ**00*, Participant	198.43			198.9			191.04		
		*N* Participant	91			91			91		
		*ICC* Participant	0.7			0.7			0.71		
		Observations	236			236			236		
		R^2^_m_/R_2__c_	0.127/743			0.145/0.747			0.157/0.757		
Anxiety subscale	Intercept		39.89	37.44 to 42.34	<0.001	39.96	37.53 to 42.4	<0.001	40.11	37.68 to 42.54	<0.001
	Session	9–1	−3.7	−5.93 to −1.47	0.001	−3.6	−5.83 to −1.38	0.002	−3.57	−5.77 to −1.38	0.002
		14–1	−11.05	−13.26 to −8.83	<0.001	−10.95	−13.17 to −8.74	<0.001	−10.53	−12.72 to −8.35	<0.001
	Condition	CBT + VR-CBT				0.13	−4.74 to 6	0.966	0.29	−5.55 to 6.14	0.922
		VR-CBT				4.58	−1.43 to 10.59	0.138	5.39	−0.64 to 11.42	0.083
	Session * Condition	9–1 * CBT + VR-CBT							−0.17	−5.26 to 4.92	0.947
		14–1 * CBT + VR-CBT							3.31	−1.75 to 8.38	0.202
		9–1 * VR-CBT							0.73	−4.83 to 6.32	0.769
		14–1 * VR-CBT							9.13	3.66 to 14.71	0.002
	Random parts	AIC	1768.02			1769.08			1764.53		
		*σ* *2*	119.3			117.5			117.2		
		*τ**00*, Participant	47.9			48			45.6		
		*N* Participant	91			91			91		
		*ICC* Participant	0.713			0.71			72		
		Observations	236			236			236		
		R^2^_m_ / R_2__c_	0.113/0.746			0.137/0.75			0.152/0.763		
	Intercept		34.31	31.74 to 36.89	<0.001	34.37	31.79 to 36.96	<0.001	34.43	31.83 to 37.02	<0.001
	Session	9–1	−4.49	−7.01 to −1.98	<0.001	−4.42	−6.94 to −1.9	<0.001	−4.55	−7.07 to −2.03	<0.001
		14–1	−12.37	−14.9 to −9.84	<0.001	−12.3	−14.84 to −9.77	<0.001	−11.98	−14.5 to −9.46	<0.001
	Condition	CBT + VR-CBT				−0.6	−6.82 to 5.63	0.951	−0.5	−6.74 to 5.73	0.874
		VR-CBT				2.83	−3.55 to 9.22	0.387	3.2	−3.24 to 9.65	0.333
	Session * Condition	9–1 * CBT + VR-CBT							−0.66	−6.5 to 5.2	0.825
Avioidance subscale		14–1 * CBT + VR-CBT							0.74	−5.18 to 6.66	0.805
		9–1 * VR-CBT							−2.19	−8.6 to 4.22	0.504
		14–1 * VR-CBT							5.9	−0.6 to 12.4	0.077
	Random parts	AIC	1798.95			1801.63			1802.6		
		*σ* *2*	128.2			128.9			129.31		
		*τ**00*, Participant	61.6			61.7			60.46		
		*N* Participant	91			91			91		
		*ICC* Participant	0.67			0.67			0.68		
		Observations	234			234			234		
		R^2^_m_/R_2__c_	0.122/0.715			0.132/0.719			0.14/0.726		

Model: random intercept; Contrasts: simple; Reference level for condition = CBT; Reference level for session = session 1.

## Data Availability

The database supporting the conclusions of this article will be made available by the authors upon request while following the institutional regulations.

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
