# Peer review of "Preliminary Study of Efficacy and Safety of Self-Administered Virtual Exposure Therapy for Social Anxiety Disorder vs. Cognitive-Behavioral Therapy"

_brainsci, 2022, doi:10.3390/brainsci12091236_

Round 1

Reviewer 1 Report (Previous Reviewer 1)

I have carefully read the revised version of the manuscript. I feel that it is suitable with an improvement concerning the Tables' layout. Specifically, Table 6 needs to be enhanced for readability and clarity. 

Author Response

Reviewer 2 Report (New Reviewer)

See attached file, please.

Author Response

Reviewer 3 Report (New Reviewer)

Dear Authors, 

Thank you for the opportunity to review the paper entitled “Preliminary study of efficacy and safety of self-administered virtual exposure therapy for social anxiety disorder vs cognitive-behavioral therapy”. The study aimed to assess the safety and efficacy of the use of a combination of CBT and VR therapy. Congratulations to the authors for addressing the topic. The paper presents a very interesting field of study, has a clear purpose, yet the clarity of the manuscript is moderate. In terms of content, apart from the obvious lack of a statistical analysis chapter, the paper presents high scientific quality. However, the editorial quality is poor - I have tried to point out minor errors for improvement. I have 5 major comments and several minor comments. Below is a list of my comments

Major: 

·       In my opinion, the introduction to the issue of SAD has been greatly simplified: 4 lines is far too short.

·       Please "set out hard" the aim of the research instead of indicating what you would like to do (line 96-98). The indicated numbered targets are a rather unusual approach. In the 1st objective besides, the same objective is described twice.

·       Please standardize the article, either use throughout the paper passive or active voice.

·       The chapter on statistical analysis is missing: please complete this chapter extensively

·       The results of Table 5, will depend on the distribution of variables, it was not mentioned that the distribution was close to normal (no statistical analysis chapter): such analysis was performed?

Minor:

·       wrong bracket in line 20

·       I suggest rewriting the abstract: instead of indicating a higher score at T2- indicate the clinical implementation (what the higher score means) and a description of the time frame of T2 (when it occurred- as a final examination?). I also suggest removing the limits from the summary - a very unusual approach

·       Line 53 and 56 lack references to meta-analyses

·       typos line 61

·       Line 64, I am not convinced about the term client-is the treated patient or home user a client?

·       I don't understand the sentence from lines 129-130. Which group is it referring to? Lines 127-130 continue to be a description of group 3?

·       I would describe the first paragram of Chapter 2 as a separate "design" subsection. 

·       Chapter 2.1 is simply Participants

·       Please use the original Consort flowchart-it is more readable and in better resolution

·       I propose to change the name of the group in line 116- in table 1 it is specified as "VR"

·       Typo line 165

·       Please add an explanation of the time-point as the note of Table 2.

·       Please standardize the headings in Table 3: order and abbreviations.

·       I suggest to have the font in table 6 reduced, it's hard to follow the results on 3 pages

·       Figure 3 are identical y-axis descriptions in both figures.

·       Line 374, does not need an explanation of the abbreviation was indicated in line 46.

·       line 416-417 is unlikely to require a separate paragraph

·       Please explain why it is indicated in the limits that the limitation is the size of the group? After all, sample size calculations were made and used, such calculations give representative results after all- that is what they are made for. Greater power could be assumed (1-β err prob)

·       The sample acknowledgement text can be removed

Author Response

Dear Reviewer,

Thank you very much for your review. I tried to take into account all your suggestions.

Best regards,

Izabela Stefaniak

PhD, Institute of Psychiatry and Neurology, Warsaw, Poland

Round 2

Reviewer 2 Report (New Reviewer)

Bearing in mind the changes conducted by authors, this ms. is accepted for publication.

Author Response

thanks

This manuscript is a resubmission of an earlier submission. The following is a list of the peer review reports and author responses from that submission.

Round 1

Reviewer 1 Report

The manuscript assessed the use of virtual reality on the treatment of social anxiety disorder. Three main groups were identified with 91 participants involved. Results evidenced the effectiveness of a virtual reality setup and a cognitive-behavioral therapy protocol. 

I found the paper interesting but feel that some issues should be addressed. Therefore, I invite the authors to tackle my concerns listed below. 

1. A more solid theoretical framework on the specific topic should be provided. A concise literature overview on the use of virtual reality setup and cognitive-behavioral protocol should be detailed. A strong rationale for the current submission should be justified. 

2. I would include eligibility criteria with both including and excluding criteria. Instaead of disqualified I wold use excluded accordingly. 

3. The heading discussions should be "Discuussion" and numbered as 4 rather than 3. 

4. Beside limitations, I suggest to include future research perspectives. 

Author Response

Dear Reviewer,

Thank you very much for your review. In this letter I will try to answer all your comments and doubts as well as to take into account all your suggestions. I explain your comments point by point. Revisions to the manuscript I marked up using the “Track Changes”

Changes from your revision are gray.

Reviewer 2 Report

TITLE

The title is direct. It would work well to give visibility to the paper.

ABSTRACT

The abstract is easy to understand. However, I think it would gain in quality if the conclusions were developed a little more (in the current version, the limitations play a leading role), and the profile of readers for whom it might be of interest were specified. On the other hand, in the Abstract the acronym "CBT" is used for the first time; therefore, it needs to be explained here.

INTRODUCTION

The introduction contains valuable information. However, I consider that it has the following shortcomings that could be improved:

·       Paragraphs should be organised to improve narrative clarity. It should be noted that each paragraph should describe one idea. However, in the current version, the first paragraph deals with very different ideas. I recommend dividing it up and working on the links between them.

·       The introduction is, in part, too much directed. I believe that, once it is better organised into paragraphs, it will be easier to complete the information to expand its theoretical framework.

·       The objective should be expressed in an academic way, in a separate paragraph (right after identifying the gap in the literature). "We would like" is not the best expression for this purpose.

MATERIALS AND METHODS

The method is literature based. I recommend addressing some possible improvements:

·       In section 2.2 (Measures), a summary table could help to get an overview of the measures used as a whole.

·       The VR system should be defined on a technical level. For example, indicate the resolution, refresh rate, field width, computer system to which it was connected, etc.

·       Regarding the virtual software, although it is not strictly necessary, it could be interesting to comment on some technical aspects. For example: which game engine was used to generate it, which hardware was used to run it in the lab and at how many FPS, and include some photographs of the simulations.

·       In section 2.3 (Treatment), Table 2 could be clearer if it incorporated an extra column to separate CBT from CBT+VR.

Typographical errors

·       Paragraphs ending without a full stop (".").

·       Sometimes a square bracket "]" is used to close a parenthesis ")".

·       Presence of double spaces.

·       Presence of blank lines within a paragraph (line 237).

·       Sometimes references are not properly commented on. For example, it is written “In one such study, that is [27]…”; when it would be clearer to the reader “In one of these studies, that of Hartanto and colleagues [27]…”.

·       In the abstract it is stated: “The objective of the study was to assess the safety and efficacy of the use of virtual reality (self-administered] vs CBT and CBT with virtual exposure to”. It might be clearer to express it as follows: “The objective of the study was to assess the safety and efficacy of the use of virtual reality (self-administered] vs CBT vs CBT with virtual exposure”.

·       Sometimes mathematical formulae are larger than the rest of the text (e.g. line 108).

DISCUSSION

An academic discussion consists of putting the results obtained in context with other studies. In the current version of the article, the first 11 lines (out of 50) focus on summarising the results. This is inappropriate. There is no need to recapitulate. It is preferable to use this space for contextualisation.

CONCLUSIONS

Including a conclusion section would help to close the narrative of the article.

Author Response

Dear Reviewer,

Thank you very much for your review. In this letter I will try to answer all your comments and doubts as well as to take into account all your suggestions. I explain your comments point by point. Revisions to the manuscript I marked up using the “Track Changes”

All changes from your revision are yellow.

Round 2

Reviewer 2 Report

I acknowledge the review work that the authors have done. They have carried out the suggested improvements, with the result that the paper has improved with respect to its original version. The document is interesting and well documented, and the research is satisfactorily developed and explained.